# Comparison of cardiac output estimates obtained from the Antares oscillometric pulse wave analysis algorithm and from Doppler transthoracic echocardiography

Alexander Stäuber[1]*, Matthias Wilhelm Hoppe[2], Harald Lapp[3], Stefan Richter[4], Marc-Alexander Ohlow[5], Marcus Dörr[6,7], Cornelia Piper[8], Siegfried Eckert[8], Michael Thomas Coll- Barroso[9], Franziska Stäuber[10], Nadine Abanador-Kamper[11,12], Johannes Baulmann[13]

1 Department of Movement and Training Science, Leipzig University, Leipzig, Germany, 2 Department of Exercise Science, Philipps University of Marburg, Marburg, Germany, 3 Department of Cardiology, Zentralklinik Bad Berka GmbH, Bad Berka, Germany, 4 Department of Cardiology, SRH Klinikum Burgenlandkreis GmbH, Naumburg, Germany, 5 Department of Cardiology, SRH Wald-Klinikum GmbH, Gera, Germany, 6 Department of Internal Medicine B, University Medicine Greifswald, Greifswald, Germany, 7 German Center for Cardiovascular Research (DZHK), Partner Site Greifswald, Greifswald, Germany, 8 Clinic for General and Interventional Cardiology/Angiology, Heart and Diabetes Center North Rhine-Westphalia, Bad Oeynhausen, Germany, 9 Department of Internal Medicine, Fliedner University of Applied Sciences, Düsseldorf, Germany, 10 Department of Sports Medicine, Chemnitz University of Technology, Chemnitz, Germany, 11 Department of Cardiology, HELIOS University Hospital Witten/Herdecke, Wuppertal, Germany, 12 Center for Clinical Medicine, Witten/Herdecke University, Wuppertal, Germany, 13 Praxis Dres. Gille/Baulmann, Rheinbach, Germany

* alexander.staeuber@gmail.com

**Data Availability Statement:** All relevant data are within the manuscript and its Supporting Information files.

## Abstract

### Background

In cardiology, cardiac output (CO) is an important parameter for assessing cardiac function. While invasive thermodilution procedures are the gold standard for CO assessment, transthoracic Doppler echocardiography (TTE) has become the established method for routine CO assessment in daily clinical practice. However, a demand persists for non-invasive approaches, including oscillometric pulse wave analysis (PWA), to enhance the accuracy of CO estimation, reduce complications associated with invasive procedures, and facilitate its application in non-intensive care settings. Here, we aimed to compare the TTE and oscillometric PWA algorithm Antares for a non-invasive estimation of CO.

### Methods

Non-invasive CO data obtained by two-dimensional TTE were compared with those from an oscillometric blood pressure device (custo med GmbH, Ottobrunn, Germany) using the integrated algorithm Antares (Redwave Medical GmbH, Jena, Germany). In total, 59 patients undergoing elective cardiac catheterization for clinical reasons (71±10 years old, 76% males) were included. Agreement between both CO measures were assessed by Bland-Altman analysis, Student's t-test, and Pearson correlations.

**Funding:** The University Medicine Greifswald and Zentralklinik Bad Berka have received non-targeted financial support by Redwave Medical GmbH for carrying out the study. The main parts of the study were financed by the internal funds of the participating hospitals. Redwave Medical GmbH had no role in study design, data collection and analysis, decision to publish, or preparation of the manuscript. The authors received no specific funding for this work. The publication of this article was funded by the Open Access Publishing Fund of Leipzig University supported by the German Research Foundation within the program Open Access Publication Funding.

**Competing interests:** A.S. has part-time obligations in Redwave Medical GmbH. M.D. has received equipment for research projects from custo med GmbH and funding for research projects from Redwave Medical GmbH. S.R. has received equipment for research projects from custo med GmbH. S.E. has received equipment for research projects from custo med GmbH. J.B. has interest in Redwave Medical GmbH, and has received equipment and lecture fees from IEM GmbH, BPLab, SMT medical GmbH & Co., SOT Medical Systems and Tensiomed. All other authors have no conflict of interest to state. Redwave Medical GmbH is patent holder for pulse wave analysis in oscillometric pulse waves that are recorded during inflation and deflation of a cuff (patent no DE 10 2017 117 337 B4) and provided information about the algorithm requested by the authors. This does not alter our adherence to PLOS ONE policies on sharing data and materials.

## Results

The mean difference in CO was 0.04 ± 1.03 l/min (95% confidence interval for the mean difference: -0.23 to 0.30 l/min) for the overall group, with lower and upper limits of agreement at -1.98 and 2.05 l/min, respectively. There was no statistically significant difference in means between both CO measures ($P = 0.785$). Statistically significant correlations between TTE and Antares CO were observed in the entire cohort (r = 0.705, $P<0.001$) as well as in female (r = 0.802, $P<0.001$) and male patients (r = 0.669, $P<0.001$).

## Conclusions

The oscillometric PWA algorithm Antares and established TTE for a non-invasive estimation of CO are highly correlated in male and female patients, with no statistically significant difference between both approaches. Future validation studies of the Antares CO are necessary before a clinical application can be considered.

## Introduction

Cardiovascular disease is a major concern worldwide due to its increasing prevalence and resulting mortality and disability rates, which also represent a large economic burden [1, 2]. In cardiology, the accurate measurement of cardiac output (CO), which is the product of stroke volume (SV) and heart rate, is of paramount importance; especially, in high-risk patients [3]. Invasive methods, such as pulmonary artery thermodilution, have traditionally been considered the gold standards for CO determination, but minimally invasive and non-invasive methods are also available, including transthoracic Doppler echocardiography (TTE) and pulse wave analysis (PWA) with non-invasive sensors [4–6].

PWA uses a mathematical analysis of the arterial blood pressure (BP) waveform to estimate CO. This waveform, which arises from a myriad of physiological elements such as left ventricular SV, aortic compliance, vascular resistance, and wave reflection phenomena, represents a complex signal in the context of arterial BP [5]. Since pulmonary artery catheterization (PAC) is a procedure only intended for certain clinical indications (e.g. assessment of patients with pulmonary hypertension, cardiogenic shock, and unexplained dyspnea) [7], TTE has become the primary imaging modality for assessing cardiac function in routine clinical settings because of its non-invasiveness, widespread use, portability, affordability, and safety (no ionized radiation) [8]. Consequently, TTE is considered as the established standard for daily clinical practice in the assessment of SV and CO [9, 10].

However, there still exists an unmet need for further non-invasive methods that can streamline the estimation of CO, minimize the complications linked to invasive approaches, and make it easier to use in non-intensive care environments. Besides TTE, numerous non-invasive devices for measuring CO have been reported in the scientific literature that utilize methods such as non-invasive PWA, pulse wave transit time, or thoracic bioimpedance [3, 11]. Nevertheless, in a recent review of available technologies for CO determination based on PWA, oscillometry was overlooked as a potential technique [4]. Importantly, oscillometry offers clear advantages, such as ease of use, rapid measurement procedure, and no need for special operator training. Potential applications encompass rapid hemodynamic evaluation in emergency rooms and outpatient settings. In this study, we aimed to compare non-invasive estimations of CO using TTE and the oscillometric PWA algorithm, Antares.

## Materials and methods

### Study population

This study was part of a multicenter study conducted between October 2017 and February 2021 for validation of a non-invasive central BP device [12, 13]. This research received approval from three Ethics Committees: Landesärztekammer Thüringen (Reg.-Nr.: 36950/2018/76), Ethics Committee University Medicine Greifswald (Reg.-Nr.: BB032/17), and Ethics Committee Ruhr University Bochum (Reg.-Nr.: 2017–219). The study strictly followed the principles outlined in the Declaration of Helsinki, and all participants provided written informed consent before their participation. Complete CO data, obtained from established TTE were available for a cohort comprising 90 adult patients. These patients had undergone elective cardiac catheterization for clinical reasons at the Heart Center of Zentralklinik Bad Berka between February 2019 and September 2020. The inclusion criteria for this study was patients aged $\geq$ 18 years with a clinical indication for cardiac catheterization. The inclusion was independent of the underlying disease or condition, as the only criteria related to the need for cardiac catheterization and ability to give informed consent. The data access for research purposes took place in the period September to October 2023. Demographic and specific clinical patient characteristics were obtained from the medical records. All 90 included patients were Caucasian of whom one patient (1.1%) was younger than 50 years, 50 patients (55.6%) were between 50 and 70 years, and 39 patients (43.3%) were older than 70 years.

### Cardiac output estimation with the Antares algorithm

The used oscillometric BP device for CO estimation was the custo screen 400 (custo med GmbH, Ottobrunn, Germany) with the integrated algorithm Antares (Redwave Medical GmbH, Jena, Germany). Redwave Medical holds a patent for PWA in oscillometric pulse waves recorded during cuff inflation and deflation (patent no DE 10 2017 117 337 B4). The Antares algorithm takes a cuff pressure signal during the deflation phase as its input, distinguishing the pulsatile signal component from the underlying cuff pressure and discerning individual pulse waves. The process involves a series of analytical steps on each pulse wave. Subsequently, grid points are pinpointed for the computation of hemodynamic parameters. The basic principle of Redwave Medical's algorithm is based on a Pulse Contour Analysis for determining the CO. For the analysis of the pulse waveform, the entire signal in the deflation process, and thus the individual pulse waves at different pressure levels, are considered. Using this signal, arterial compliance is determined as an important input to the algorithm. Each of these pulse waves has an individual shape–the Pulse Wave Phenotype®. Subsequently, the grid points at individual pulse waves, and thus the phenotype, are determined. For CO estimation with Antares, the grid points had to meet the criteria of being both detectable and sufficiently stable within a specified region of interest in the signal, while also exhibiting low variance (Fig 1).

Pulse wave signals that do not meet these criteria are automatically recognized and rejected by the algorithm. In this context, the algorithm was unable to determine CO in 34% cases due to inappropriate data. The remaining 59 patients had a pulse waveform with clearly identifiable and stable grid points, which is a basic requirement for meaningful calculations in PWA [4].

For each patient, the estimated CO derived from brachial oscillometric BP measurement on the left upper arm at the end of the cardiac catheterization procedure was utilized to compare with the CO determined by TTE. For a more detailed description of the methodological approach of the BP measurement procedure, please refer to Dörr et al. [12].

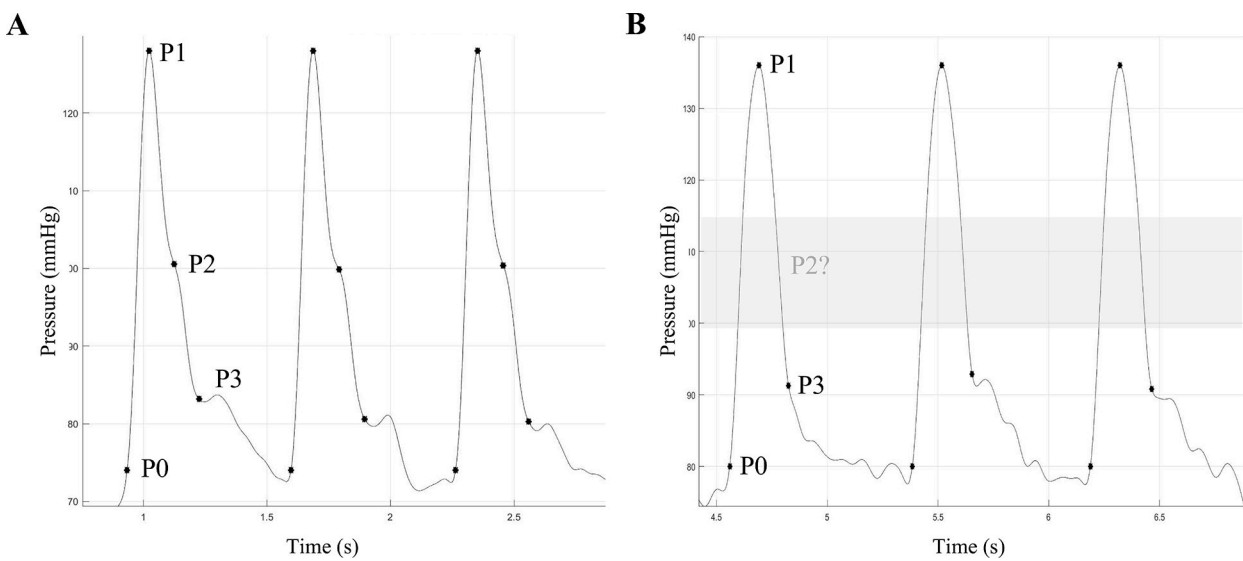

**Fig 1.** Example of extracted oscillometric pulse waves of (A) an 84-year-old female patient with detectable and stable grid points (P0-P3) and (B) a 51-year-old male patient with non-detectable grid point P2. P0-P3 are necessary for the non-invasive estimation of cardiac output by the Antares algorithm for pulse wave analysis.

### Cardiac output estimation with TTE

CO was determined using two-dimensional TTE with the cardiac ultrasound systems EPIQ CVx (Philips Healthcare, Andover, MA, USA), VIVID S70 and E95 (GE HealthCare GmbH, Solingen, Germany) as a standardized routine examination procedure with a median time difference of one day before cardiac catheterization. All TTEs have been performed by experienced cardiologists according to established TTE guidelines [14, 15]. For each included subject, CO was calculated using the TOMTEC Arena software using the data acquired by Philips EPIQ 7 echo units (Philips Medical Systems). The diameter of the left ventricular outflow tract (LVOT) was assessed in zoomed parasternal long-axis views during early systole at the level of the aortic cusp insertion (aortic annulus). The LVOT velocity-time integral was captured from the apical 5-chamber view, with the sample volume positioned approximately 5 mm proximal to the aortic valve. Filters were fine-tuned to ensure precise visualization of the pulsed-wave Doppler signal and the closure click of the aortic valve. Three measurements were taken and averaged for both the LVOT diameter and velocity-time integral. The heart rate value used for cardiac output (CO) calculation was extracted from the pulsed-wave Doppler recording of the LVOT velocity-time integral. Measurements of left ventricular end-diastolic and end-systolic diameters were conducted in M-mode parasternal long-axis views, situated 1 cm below the mitral annulus. This was achieved either with the cursor perpendicular to the long axis of the left ventricle or through two-dimensional echocardiography. Left ventricular ejection fraction was determined using the Simpson biplane method [15] or, in cases of poor acoustic window, through visual estimation. Stroke volume (SV) was calculated using the formula: $SV = (\pi \times LVOT\ diameter^2/4) \times LVOT$ velocity-time integral, and CO was obtained by multiplying SV by the heart rate.

Both CO measures (oscillometric PWA and TTE) were also normalized to body surface area (BSA), using the Mosteller formula [16] and expressed as cardiac index (CI = CO/BSA).

All data collections were performed with the subject in an undisturbed resting phase in the supine position, free from acute hemodynamic interventions, free from acute medication changes, and not conversing.

## Statistics

The normal distribution of continuous data and mean differences was assessed using the Kolmogorov-Smirnov test. According to data distribution, differences in means between hemodynamic data within the total group, the female, and the male group were assessed using paired Student's t-test. Differences in means between both sexes were analyzed using the unpaired Student's t-test. Agreement between TTE and Antares CO was evaluated using Bland-Altman plots with limits of agreement (±1.96 SD) and their 95% confidence intervals [17]. The Bravais-Pearson correlation coefficient was used to assess the strength and direction of linear association between the TTE and Antares CO. Based on the conventional definition, a correlation coefficient between 0.40–0.69 and 0.70–0.89 was interpreted as moderate and strong, respectively [18]. In addition, scatter plots were created for a graphical overview. A two-sided P value of less than 0.05 was considered statistically significant in all analyses. All statistical calculations were done using SPSS version 22 (IBM Corp, Armonk, New York, USA).

## Results

Table 1 shows the clinical and demographic characteristics of the study population.

Regarding the hemodynamic data collected using TTE, cardiac catheterization (aortic BP), and the oscillometric brachial BP device, the statistical analysis showed no statistically significant differences among the entire group, female, and male patients, or between the sexes, except for lower aortic diastolic BP ($P = 0.044$) and higher left ventricular ejection fraction ($P = 0.035$) observed in the female group (Table 2).

For CO measured with TTE, significant correlations with Antares CO have been found for the total group (r = 0.705, $P<0.001$) as well as for female (r = 0.802, $P<0.001$) and male patients (r = 0.669, $P<0.001$). Findings for CO are presented for the total group in Fig 2. The mean of differences for CO was 0.04 ± 1.03 l/min (95%CI for mean of differences: -0.23 to 0.30 l/min) for the total group, with a lower and upper limit of agreement of -1.98 and 2.05 l/min, respectively. The 95% confidence interval for the lower limit of agreement was -2.44 to -1.51 l/min and 1.59 to 2.52 l/min for the upper limit of agreement. In addition, the mean of

**Table 1. Patient characteristics.**

|  | Total | Females | Males |
|---|---|---|---|
| Patients, n | 59 | 14 (24%) | 45 (76%) |
| Age, years | 71.3 ± 10.3 | 74.9 ± 9.0 | 70.2 ± 10.4 |
| Weight, kg | 85.9 ± 15.3 | 74.4 ± 8.2 | 89.5 ± 15.3 |
| Height, cm | 172.3 ± 10.2 | 159.9 ± 6.2 | 176.1 ± 7.8 |
| BMI, kg/m$^2$ | 28.9 ± 4.2 | 29.2 ± 4.0 | 28.8 ± 4.3 |
| Body surface area, m$^2$ | 2.0 ± 0.2 | 1.8 ± 0.1 | 2.1 ± 0.2 |
| Arterial hypertension | 55 (93%) | 12 (86%) | 43 (96%) |
| Dyslipidaemia | 27 (46%) | 6 (43%) | 21 (47%) |
| Diabetes mellitus | 27 (46%) | 6 (43%) | 21 (47%) |
| Chronic kidney disease | 9 (15%) | 2 (14%) | 7 (16%) |
| Prior stroke | 6 (10%) | 2 (14%) | 4 (9%) |
| Prior myocardial infarction | 11 (19%) | 0 (0%) | 11 (24%) |
| Patients undergoing PCI | 28 (47%) | 7 (50%) | 21 (47%) |
| Chronic heart failure | 11 (19%) | 0 (0%) | 11 (24%) |
| Coronary heart disease | 34 (58%) | 7 (50%) | 27 (60%) |

Continuous parameters presented as mean ± SD, categorial variables presented as number of subjects (%). PCI, percutaneous coronary intervention

**Table 2. Hemodynamic parameters obtained with custo screen 400 device with integrated oscillometric pulse wave analysis algorithm Antares and transthoracic doppler echocardiography (TTE) of total group and stratified by sex.**

| | Total | Females | Males | Females vs. males P value |
|---|---|---|---|---|
| Patients, n | 59 | 14 | 45 | |
| Brachial systolic BP, mmHg | 140.4 ± 22.8 | 138.1 ± 24.1 | 141.2 ± 22.7 | 0.660 |
| Brachial diastolic BP, mmHg | 80.2 ± 10.8 | 76.8 ± 10.0 | 81.2 ± 10.9 | 0.180 |
| Brachial mean arterial pressure, mmHg | 101.3 ± 15.0 | 99.6 ± 16.9 | 101.8 ± 14.5 | 0.649 |
| Aortal systolic BP, mmHg | 132.3 ± 24.0 | 128.3 ± 30.5 | 133.5 ± 21.9 | 0.893 |
| Aortal diastolic BP, mmHg | 67.7 ± 12.1 | 60.3 ± 12.1 | 70.0 ± 11.2 | **0.044** |
| Aortal mean arterial pressure, mmHg | 93.8 ± 15.0 | 87.7 ± 18.1 | 95.7 ± 13.6 | 0.264 |
| Heart rate custo screen 400, bpm | 69.8 ± 12.5 ($P = 0.140$)[#] | 69.1 ± 14.8 ($P = 0.071$)[#] | 70.1 ± 11.9 ($P = 0.237$)[#] | 0.798 |
| Heart rate TTE, bpm | 70.2 ± 12.8 | 69.6 ± 14.5 | 70.4 ± 12.4 | 0.535 |
| LVEF, % | 52.1 ± 12.6 | 58.3 ± 12.9 | 50.2 ± 12.0 | **0.035** |
| Cardiac output TTE, l/min | 4.13 ± 1.44 ($P = 0.785$)[#] | 3.84 ± 1.95 ($P = 0.678$)[#] | 4.22 ± 1.25 ($P = 0.993$)[#] | 0.402 |
| Cardiac output Antares, l/min | 4.16 ± 0.89 | 3.99 ± 0.93 | 4.22 ± 0.88 | 0.414 |
| Cardiac index TTE, l/min/m$^2$ | 2.04 ± 0.70 ($P = 0.639$)[#] | 2.12 ± 1.06 ($P = 0.665$)[#] | 2.02 ± 0.56 ($P = 0.820$)[#] | 0.637 |
| Cardiac index Antares, l/min/m$^2$ | 2.07 ± 0.44 | 2.20 ± 0.50 | 2.03 ± 0.41 | 0.199 |

Continuous parameters presented as mean ± SD, categorial variables presented as number of subjects (%). BP, blood pressure; bpm, beats per minute; LVEF, left ventricular ejection fraction; TTE, transthoracic Doppler echocardiography.

[#]Comparison of measurements by device

differences was 0.15 ± 1.32 for female patients and 0.00 ± 0.93 for male patients. The scatter plots and Bland-Altman plots with additional information for female and male patients can be derived from Figs 3 and 4.

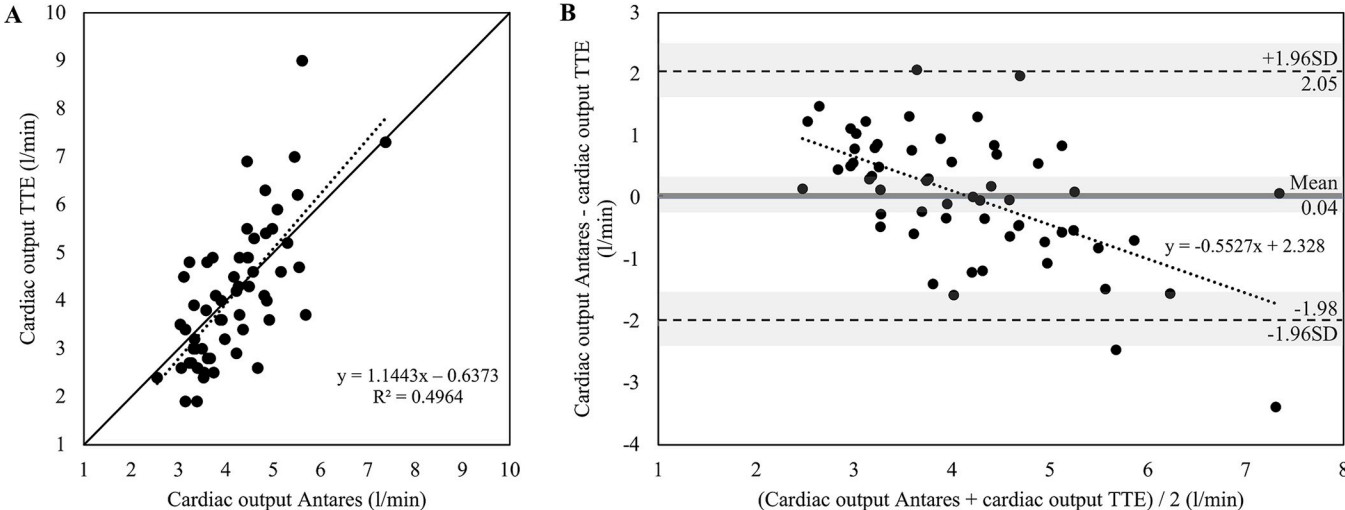

**Fig 2. Relationship between cardiac output (CO) measured with transthoracic Doppler echocardiography (TTE) and estimated with the Antares oscillometric pulse wave analysis algorithm in patients with cardiovascular diseases (n = 59).** A) Scatter plot for TTE CO and Antares CO. Black dotted line: linear regression line. $R^2$, coefficient of determination. Pearson's r = 0.705, $P<0.001$. Black line: identity line. B) Bland-Altman plot for TTE CO and Antares CO with the representation of mean difference (grey line), limits of agreement (black dashed line) from ±1.96 SD and corresponding 95% confidence intervals (gray shaded areas). Mean difference ± SD: 0.04 ± 1.03 l/min. Black dotted line: linear regression line.

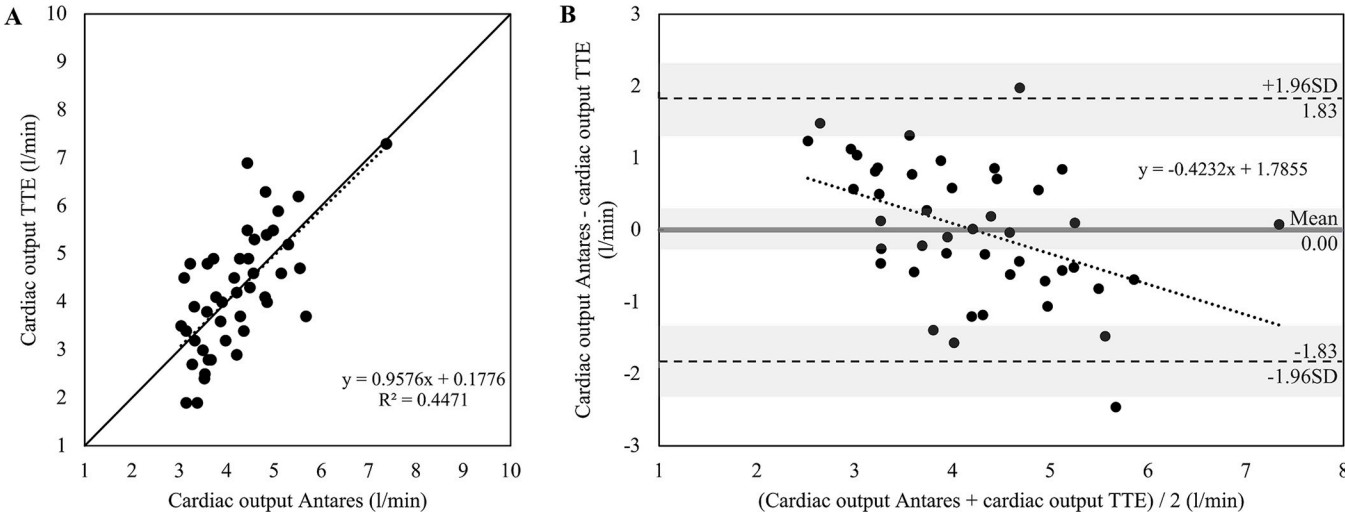

**Fig 3. Relationship between cardiac output (CO) measured with transthoracic Doppler echocardiography (TTE) and estimated with the Antares oscillometric pulse wave analysis algorithm in male patients with cardiovascular diseases (n = 45).** A) Scatter plot for TTE CO and Antares CO. Black dotted line: linear regression line. $R^2$, coefficient of determination. Pearson's r = 0.669, $P$<0.001. Black line: identity line. B) Bland-Altman plot for TTE CO and Antares CO with the representation of mean difference (grey line), limits of agreement (black dashed line) from ±1.96 SD and corresponding 95% confidence intervals (gray shaded areas). Mean difference ± SD: 0.00 ± 0.93 l/min. Black dotted line: linear regression line.

## Discussion

In this study, we aimed to compare TTE and the Antares oscillometric PWA algorithm for a non-invasive estimation of CO. Our findings revealed a strong correlation between CO estimations obtained through Antares and those derived from TTE. This correlation was consistent among both female and male patients, and we observed a good level of agreement (accuracy) between the two methods, with no statistically significant bias detected.

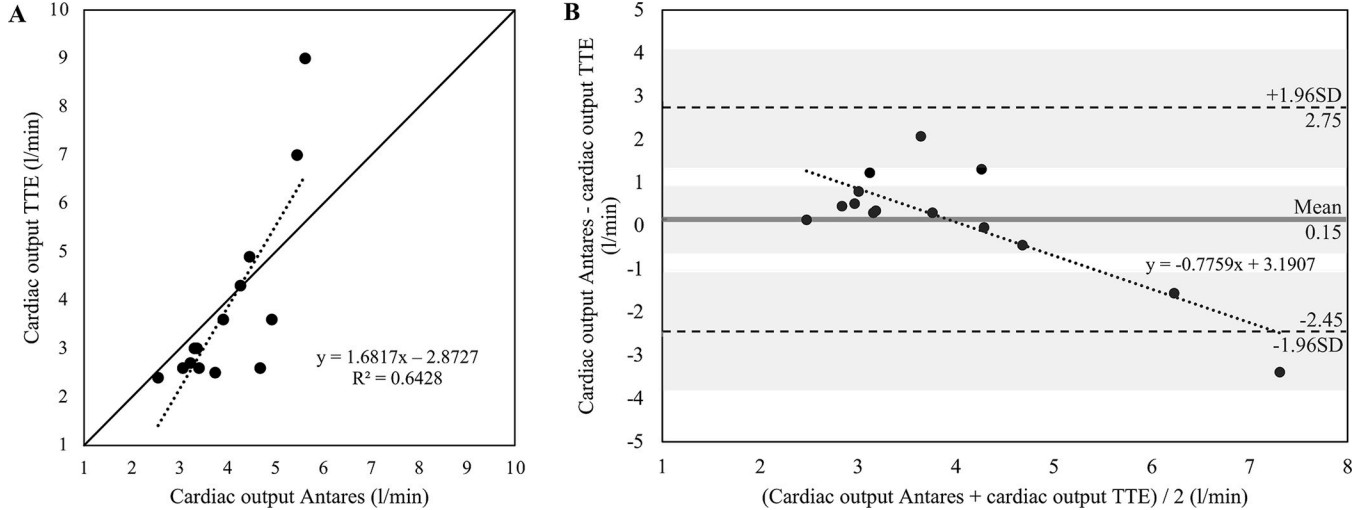

**Fig 4. Relationship between cardiac output (CO) measured with transthoracic Doppler echocardiography (TTE) and estimated with the Antares oscillometric pulse wave analysis algorithm in female patients with cardiovascular diseases (n = 14).** A) Scatter plot for TTE CO and Antares CO. Black dotted line: linear regression line. $R^2$, coefficient of determination. Pearson's r = 0.802, $P$<0.001. Black line: identity line. B) Bland-Altman plot for TTE CO and Antares CO with the representation of mean difference (grey line), limits of agreement (black dashed line) from ±1.96 SD and corresponding 95% confidence intervals (gray shaded areas). Mean difference ± SD: 0.15 ± 1.32 l/min. Black dotted line: linear regression line.

To our knowledge, this is the first study investigating the comparability between a cuff-based oscillometric PWA-derived CO with the TTE-derived CO in patients with various cardiovascular disease. At this point, it should be mentioned that the present study does not claim to be a validation study. While the research results emphasize the comparability of both CO determination methods, it is important to consider the specific constraints associated with oscillometric PWA and TTE. A crucial aspect of oscillometric PWA, including the Antares algorithm, is its dependence on an optimal arterial pressure signal, which may be vulnerable to interference in specific clinical situations [4, 19]. Thus, low oscillations in the context of severe arterial hypotension and shock and susceptibility to oscillation fluctuations in the context of tachycardic arrhythmias, extrasystoles, and movement artefacts in rhythmic oscillations (e.g., muscle tremor, transport, etc.) could be interfering factors. Therefore, the usage of oscillometric PWA may be limited to selected patients with adequate signal quality in non-intensive care environments. Additionally, external factors such as positioning (at heart level), selection (cuff size), and application of the upper arm cuff (correct fit, tightness) can also affect the pulse wave signals and thus compromise the accuracy and precision of the algorithm. Furthermore, the effectiveness of an oscillometric PWA algorithm such as Antares relies on assumptions about vascular physiology or hemodynamic principles that may not apply to all patient cohorts or clinical scenarios. In the present study, inappropriate oscillometry-based pulse wave data without clearly identifiable and stable grid points led to an automated algorithmic exclusion of 34% of the examined patients. For a clinical application, this still poses a significant challenge that needs to be solved in the future.

Described limitations of TTE-derived CO include operator dependency and difficulty in assessing patients with large body habitus or recent chest surgery or trauma [8]. Additionally, TTE might encounter limitations due to suboptimal image quality (such as a poor acoustic window), difficulties in visualizing left ventricular substructures, challenges in detecting regional wall motion abnormalities, constraints in quantitative assessment, and temporal restrictions. Of note, the limiting factors already described here for oscillometry also lead to difficulties when determining CO using TTE. Nevertheless, this method can be considered an acceptable reference for CO estimation in certain patient groups, and its use can be justified by its safety, reproducibility and reasonably accurate results, especially given its widespread use in routine clinical practice [10, 20].

In comparison to TTE-derived CO normative reference values involving 4040 white European adults with normal BP and no history of cardiovascular disease [10], both male and female patients in our study exhibited, on average, reduced CO values obtained with TTE. Since CO is clearly influenced by gender and body size, Rusinaru et al [10] proposed CI as a highly stable and therefore better parameter for the assessment of a low output state that is independent of these factors. The authors defined a low output state by a CI of $<1.9\,\text{l/min/m}^2$ as measured by TTE. In our study, 46% of the patients had a CI below this value. However, these low CO and CI values may be because the patients had underlying cardiovascular conditions with a clinical indication for cardiac catheterization. Nevertheless, the Antares algorithm was able to map the low CO and CI values with sufficient accuracy, although a positive linear trend of mean differences proportional to the magnitude of CO values can be seen in our data. Compared with other studies that investigated the accuracy of an oscillometric based algorithm for CO estimation in intensive care patients, Papaioannou et al. [21] and Reshetnik et al. [22] found a higher mean bias for CO of -1.12 (LOA: -3.81 to 1.58) and 1.3 (LOA: -3.5 to 6.1) l/min, respectively. Interestingly, a positive trend in the differences proportional to the magnitude of the CO and CI measures can also be seen in these studies. However, a direct comparison and interpretation of these results may not be appropriate, as the transpulmonary thermodilution method was used as method for CO determination. In addition, in the present

study, the Antares oscillometric PWA algorithm was not operable in patients with ambiguous or unclear grid points for their pulse waves. Hence, a direct comparison of the study results is difficult.

Another finding of our study is the aspect of different correlations between oscillometric PWA algorithm and TTE in female and male patients. The differences observed could be attributed to a variety of factors. These include sex differences in vascular structure and function and hormonal influences on cardiovascular dynamics [23]. Additionally, there are disparities in heart size, with females typically having smaller hearts than males, and variations in heart function, such as higher left ventricular ejection fraction in females compared to males and higher CO in males compared to females [24]. Differences in body composition [25] (which may affect the transmission of pressure waves), potential differences in underlying cardiovascular disease between the sexes, and the possibility of sampling bias may also contribute to these observed differences. Therefore, further research is needed to gain a deeper understanding of these sex-specific differences and the underlying mechanisms, with the future aim of developing an algorithm that provide comparable performance in both sexes.

We acknowledge several limitations of our study. Firstly, no test-retest protocol for TTE and algorithm-derived CO determination was carried out in this study, which does not allow statistical conclusions to be made about precision, as recommended by Critchley and Critchley [26] and Cecconi et al. [27]. Secondly, the issue on inappropriate oscillometric pulse wave data may limit the algorithm's applicability to a broader patient population with less-than-ideal data quality. As a result, the feasibility of the Antares algorithm as an alternative to TEE in the clinical setting is currently limited. Nonetheless, the algorithm automatically rejected pulse waves from affected cases due to the high variance of grid points or their undetectability. Given that the Antares algorithm is capable of rejecting data that would make a valid analysis impossible, it can prevent false outputs. For medical applications, however, it is recommended that the algorithm be further optimized to be able to robustly process these cases as well. Subsequent studies should then explore the algorithm's performance in real-world scenarios with diverse patient profiles to define and ensure the possibilities of clinical utility. Further limitations are the low number of female study participants, who made up only 24% of the study population, and the high average age of the patients (71 years). Future studies should take these limitations into account and use more accurate methods for CO determination (e.g., PAC, cardiac MRI) as part of a comprehensive validation.

## Conclusions

This study demonstrated that the non-invasive estimation of CO by the Antares oscillometric PWA algorithm and by established TTE are highly correlated in male and female patients, with no statistically significant difference between both approaches. The drop-out due to the grid point issue in the pulse wave signals is a challenge that needs to be overcome. Additionally, future validation studies of the Antares CO are necessary before a clinical application can be considered.

## Supporting information

**S1 Dataset.**
(XLSX)

## Acknowledgments

The authors thank Mrs. Andrea Rosenthal for support in data handling and management.

## Author Contributions

**Conceptualization:** Johannes Baulmann.

**Formal analysis:** Alexander Stäuber.

**Investigation:** Harald Lapp, Stefan Richter, Marc-Alexander Ohlow, Marcus Dörr, Siegfried Eckert.

**Methodology:** Johannes Baulmann.

**Project administration:** Marcus Dörr.

**Supervision:** Johannes Baulmann.

**Writing – original draft:** Alexander Stäuber.

**Writing – review & editing:** Matthias Wilhelm Hoppe, Harald Lapp, Stefan Richter, Marc-Alexander Ohlow, Marcus Dörr, Cornelia Piper, Siegfried Eckert, Michael Thomas Coll-Barroso, Franziska Stäuber, Nadine Abanador-Kamper, Johannes Baulmann.

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
