## [Decision Letter · Decision Letter 0]

13 Mar 2024

PONE-D-24-03078Comparison of cardiac output estimates obtained from the Antares oscillometric pulse wave analysis algorithm and from Doppler transthoracic echocardiographyPLOS ONE

Dear Dr. Stäuber,

Thank you for submitting your manuscript to PLOS ONE. After careful consideration, we feel that it has merit but does not fully meet PLOS ONE’s publication criteria as it currently stands. Therefore, we invite you to submit a revised version of the manuscript that addresses the points raised during the review process.

We look forward to receiving your revised manuscript.

Kind regards,

Giacomo Pucci

Academic Editor

PLOS ONE

Reviewers' comments:

Reviewer's Responses to Questions

**Comments to the Author**

1. Is the manuscript technically sound, and do the data support the conclusions?

Reviewer #1: Yes

2. Has the statistical analysis been performed appropriately and rigorously? 

Reviewer #1: Yes

3. Have the authors made all data underlying the findings in their manuscript fully available?

Reviewer #1: Yes

4. Is the manuscript presented in an intelligible fashion and written in standard English?

Reviewer #1: Yes

5. Review Comments to the Author

Reviewer #1: In this very interesting article, the authors evaluated the agreement/correlation of the Antares oscillometric pulse wave analysis algorithm and Doppler transthoracic echocardiography (TTEcho) for cardiac output measurements in patients undergoing elective cardiac catheterization for clinical reasons. The authors demonstrate that Antares oscillometric pulse wave analysis algorithm measurements are comparable to measurements by TTEcho.

#1 - Although the authors observed a strong correlation and a good agreement between the methods, the Antares oscillometric pulse wave analysis algorithm could not determine cardiac output in 34% of cases due to inappropriate data. Can the authors explain why the algorithm could not determine cardiac output in these patients? Could this limit the feasibility of using the algorithm as an alternative to TTEcho in any clinical setting?

#2 - The authors should specify the sampling method and how the sample size was defined in the Methods.

#3 - The reasons for the elective cardiac catheterization should be present in the Results.

#4 - In the Results, the authors stated, "Regarding the hemodynamic data collected using TTE, cardiac catheterization (aortic BP), and the oscillometric brachial BP device, the statistical analysis showed no statistically significant differences among the entire group, female, and male patients, or between the sexes, except for a significantly higher aortic diastolic BP observed in the female group (Table 2)". However, the LVEF was lower in males than females (p = 0.035).

#5 - In the Discussion, the authors should mention the inherent limitations of the Antares oscillometric pulse wave analysis algorithm and TTEcho to ensure readers understand the methods' limitations.

#6 - Is it possible to explain the best correlations between the Antares oscillometric pulse wave analysis algorithm in men rather than women?

6. PLOS authors have the option to publish the peer review history of their article (what does this mean?). If published, this will include your full peer review and any attached files.

Reviewer #1: No

---

## [Author Response · Author response to Decision Letter 0]

20 Mar 2024

Dear Prof. Pucci, dear reviewer, 

Thank you for your valuable comments and advice, which helped us to improve the article significantly. 

Please refer to the "Response to Reviewers" letter for a detailed response. 

Best regards

Alexander Stäuber

---

## [Decision Letter · Decision Letter 1]

12 Apr 2024

Comparison of cardiac output estimates obtained from the Antares oscillometric pulse wave analysis algorithm and from Doppler transthoracic echocardiography

PONE-D-24-03078R1

Dear Dr. Stäuber,

We’re pleased to inform you that your manuscript has been judged scientifically suitable for publication and will be formally accepted for publication once it meets all outstanding technical requirements.

Kind regards,

Giacomo Pucci

Academic Editor

PLOS ONE

Reviewers' comments:

Reviewer's Responses to Questions

**Comments to the Author**

1. If the authors have adequately addressed your comments raised in a previous round of review and you feel that this manuscript is now acceptable for publication, you may indicate that here to bypass the “Comments to the Author” section, enter your conflict of interest statement in the “Confidential to Editor” section, and submit your "Accept" recommendation.

Reviewer #1: All comments have been addressed

2. Is the manuscript technically sound, and do the data support the conclusions?

Reviewer #1: Yes

3. Has the statistical analysis been performed appropriately and rigorously? 

Reviewer #1: Yes

4. Have the authors made all data underlying the findings in their manuscript fully available?

Reviewer #1: Yes

5. Is the manuscript presented in an intelligible fashion and written in standard English?

Reviewer #1: Yes

6. Review Comments to the Author

Reviewer #1: The authors have made appropriate adjustments to the original submission. I have no further recommendations.

---

## [Editor Report · Acceptance letter]

30 Apr 2024

PONE-D-24-03078R1 

PLOS ONE

Dear Dr. Stäuber, 

I'm pleased to inform you that your manuscript has been deemed suitable for publication in PLOS ONE. Congratulations! Your manuscript is now being handed over to our production team.

Kind regards, 

on behalf of

Dr. Giacomo Pucci 

Academic Editor

PLOS ONE